# The Role of Kidney Function in Predicting COVID-19 Severity and Clinical Outcomes: A Retrospective Analysis

**DOI:** 10.3390/idr17040079

**Published:** 2025-07-07

**Authors:** Victor Muniz de Freitas, Érika Bevilaqua Rangel

**Affiliations:** 1Department of Medicine, Nephrology Division, Federal University of São Paulo, Borges Lagoa Street, 783, 6th Floor, Vila Clementino, São Paulo 04038-031, SP, Brazil; victor.muniz.freitas@gmail.com; 2Instituto Israelita de Ensino e Pesquisa Albert Einstein, Hospital Israelita Albert Einstein, São Paulo 05652-900, SP, Brazil

**Keywords:** COVID-19, eGFR, vital signs, laboratory parameters, outcomes

## Abstract

**Background:** Coronavirus disease 2019 (COVID-19) involves a complex interplay of dysregulated immune responses, a pro-inflammatory cytokine storm, endothelial injury, and thrombotic complications. This study aimed to evaluate the impact of kidney function on clinical, laboratory, and outcome parameters in patients hospitalized with COVID-19. **Methods:** We conducted a retrospective analysis of 359 patients admitted during the first wave of COVID-19, stratified by estimated glomerular filtration rate (eGFR < 60 vs. ≥60 mL/min/1.73 m^2^). Data on demographics, vital signs, laboratory values, and clinical outcomes—including mortality, hemodialysis requirement, intensive care unit (ICU) admission, and mechanical ventilation (MV)—were collected. Univariate and multivariate linear regression, as well as area under the receiver operating characteristic curve (AUC-ROC) analyses, were performed. A *p*-value < 0.05 was considered statistically significant. **Results:** Patients with an eGFR < 60 were older and more likely to have systemic hypertension, chronic kidney disease, a history of solid organ transplantation, and immunosuppressive therapy. This group showed higher rates of mortality (41.6% vs. 19.2%), hemodialysis requirement (32.3% vs. 9.6%), ICU admission (50.9% vs. 37.9%), and MV (39.8% vs. 21.2%). Laboratory results revealed acidosis, anemia, lymphopenia, elevated inflammatory markers, and hyperkalemia. **Conclusions:** An admission eGFR < 60 mL/min/1.73 m^2^ is associated with worse clinical outcomes in COVID-19 and may serve as a simple, early marker for risk stratification.

## 1. Introduction

Coronavirus disease 2019 (COVID-19), caused by the Severe Acute Respiratory Syndrome Coronavirus-2 (SARS-CoV-2), has resulted in significant global mortality. The pandemic has led to approximately 120 deaths per 100,000 people worldwide, with excess mortality exceeding 300 deaths per 100,000 in 21 countries [1].

The pathophysiology of COVID-19 is multifaceted and includes a dysregulated immune response, a pro-inflammatory cytokine storm, endothelial injury, and systemic thrombotic events [2]. Although the lungs are the primary site of infection, these systemic processes also affect other organs, including the kidneys, brain, and heart.

A comprehensive review has identified several risk factors for COVID-19 progression and mortality. These include demographic factors (age, male sex, and smoking), preexisting conditions (e.g., cardiovascular and cerebrovascular diseases, chronic kidney disease [CKD], hypertension, diabetes, cancer, and chronic obstructive pulmonary disease), vital signals and symptoms (e.g., respiratory distress, hypotension, hypoxemia, tachycardia, hemoptysis, abdominal pain, anorexia, fatigue, fever, myalgia, and arthralgia), and laboratory abnormalities (e.g., elevated lactate, C-reactive protein, D-dimer, procalcitonin, lactate dehydrogenase, blood white cell and neutrophil counts, myocardial injury markers, B-type natriuretic peptide, aspartate aminotransferase, bilirubin, interleukin-6, creatinine, urea, creatinine kinase, and erythrocyte sedimentation rate, and reduced platelet and lymphocyte counts and albumin levels), radiological findings (evidence of consolidative infiltrate and pleural effusion), and a high sequential organ failure assessment (SOFA) score [3,4,5,6,7,8,9]. These findings underscore the systemic nature of COVID-19 and its potential for widespread complications.

Despite being primarily a respiratory illness marked by severe pulmonary involvement, COVID-19 has demonstrated a multisystemic nature that further burdens already overwhelmed healthcare systems. Among its numerous systemic manifestations, including sepsis and thromboembolic events [10,11,12], renal impairment directly linked to SARS-CoV-2 is particularly significant, given the high prevalence of preexisting kidney disease and comorbidities that compromise kidney function.

During the first wave of the pandemic, acute kidney injury (AKI) was reported in both previously healthy individuals and those with preexisting conditions, with etiologies ranging from pre-renal hypoperfusion and direct viral injury to cytokine-mediated damage [13,14,15].

Notably, the prognostic value of kidney function—especially estimated glomerular filtration rate (eGFR)—at hospital admission has not been fully elucidated, particularly in patients without known CKD. To address this gap, we conducted a retrospective analysis to evaluate the association between admission eGFR and clinical outcomes in patients hospitalized during the first wave of the pandemic at a tertiary care center.

## 2. Materials and Methods

### 2.1. Study Population

This retrospective study analyzed the electronic health records of all patients admitted to Hospital São Paulo, the university hospital of Escola Paulista de Medicina—UNIFESP, in São Paulo, Brazil, between March and August 2020. All patients presented with respiratory syndrome during the first wave of the COVID-19 pandemic.

Patients arriving at the Emergency Department with respiratory symptoms were triaged to a dedicated Respiratory Infection Unit within the Emergency Room. A standardized assessment form, previously described by our group [8,9], was used to collect data on symptoms, comorbidities, and vital signs at admission.

Inclusion criteria were: (1) age ≥ 18 years, (2) hospitalization at Hospital São Paulo, and (3) complete outcome data. Exclusion criteria included pregnancy, transfer to another hospital, absence of respiratory syndrome due to COVID-19 at admission, and missing data on mechanical ventilation or hemodialysis.

### 2.2. Assessment

Data were extracted from admission forms and electronic medical records, including demographic characteristics, admission/discharge dates, comorbidities, symptoms, laboratory values, and outcomes (intensive care unit [ICU] admission, mechanical ventilation [MV], hemodialysis, and mortality).

Systemic arterial hypertension (SAH) was defined as the use of antihypertensive medications. Diabetes mellitus (DM) was identified as the use of insulin and/or oral hypoglycemic agents. Cardiovascular disease included documented heart failure and/or coronary artery disease. Body mass index was classified according to World Health Organization guidelines: <25 kg/m^2^ (normal), ≥25–29.9 kg/m^2^ (overweight), and ≥30 kg/m^2^ (obesity). CKD was defined as an estimated glomerular filtration rate (eGFR) < 60 mL/min/1.73 m^2^ for at least three months, or evidence of kidney damage based on urinalysis or imaging, according to historical medical records. Diagnoses were coded using the International Classification of Diseases-10.

Due to limited data availability during the pandemic, admission eGFR < 60 mL/min/1.73 m^2^ was used as a surrogate for impaired kidney function, recognizing that this could represent undiagnosed CKD or acute kidney injury (AKI).

Calculated variables included the shock index, defined as $\frac{Heart\;Rate\;(HR)}{systolic\;blood\;pressure\;(SBP)}$ [16] and the ROX Index (Respiratory Rate Oxygenation Index), calculated as $\frac{Pulse\;oximetry\;(SpO_{2})}{Fraction\;of\;inspired\;oxygen\;(FiO_{2})\times respiratory\;rate\;(RR)}$ [17]. The ROX was calculated assuming an FiO_2_ of 21% (ambient air), as vital signs were recorded before oxygen supplementation. eGFR was calculated using the CKD-EPI 2021 creatinine equation [18].

This study was approved by the Ethics Committee of the Federal University of São Paulo (CAAE: 41400720.7.0000.5505, approved on 6 February 2021). All procedures followed institutional and national guidelines, consistent with the Declaration of Helsinki. The requirement for informed consent was waived due to the use of anonymized data.

### 2.3. Statistical Analysis

Patients were categorized into two groups based on eGFR at admission: <60 and ≥60 mL/min/1.73m^2^.

The primary objective was to evaluate the association between admission eGFR and demographic, clinical, and laboratory variables. Secondary objectives included assessing the relationship between eGFR and adverse clinical outcomes (mortality, need for hemodialysis, ICU admission, need for MV).

Continuous variables are reported as median and interquartile range [Q1; Q3] due to non-normal distributions (confirmed by the Shapiro–Wilk normality test; see Appendix A). eGFR is presented as mean ± standard deviation [SD]. Categorical variables are expressed as percentages.

Univariate linear regression was conducted to explore associations, reporting adjusted R^2^ (coefficient of determination), F-statistics, and *p*-values. Adjusted R^2^ accounts for the number of predictors in a model and reflects the proportion of variation in the dependent variable explained by the independent variables, while correcting for potential overfitting. The F-test statistic compares the variance explained by the model to the variance not explained (residual) variance; a higher F-statistic suggests that the model accounts for a significant portion of the total variance.

Variables with a *p*-value < 0.1 in univariate analyses were included in the multivariate models. Model performance was assessed using adjusted R^2^, which indicates the proportion of variance in the outcome explained by the model while accounting for the number of predictors. Predicted values from these models were then used to evaluate how well they distinguished between patients who experienced each outcome (mortality, hemodialysis, ICU, and MV) and those who did not. To assess the diagnostic ability of these models and variables, we performed Receiver Operating Characteristic (ROC) curve analyses, comparing sensitivity (true positive rate) and 1-specificity (false positive rate). The Area Under the Curve (AUC) quantified the model’s overall discriminatory power. To compare AUCs between ROC curves (multivariate model vs. eGFR alone), we used DeLong’s test. This test determines whether the difference in predictive performance is statistically significant. We also calculated the Z-score for the AUC difference. A non-significant result (*p* > 0.05) suggests similar predictive accuracy between models, while a significant result (*p* < 0.05) indicates that one model is statistically superior.

Statistical analyses were performed using jamovi software (v. 2.6.13, Jamovi, Sydney, Australia) and the following packages: flexplot, jsq, jsurvival, meddecide, snowCluster, ClinicoPathDescriptives, and Dispersion. Variables with >30% missing data were excluded. A significance level of 5% (α = 0.05) was adopted for all tests.

## 3. Results

We analyzed data from 359 hospitalized patients with COVID-19. The overall mean eGFR was 65.3 ± 34.3 mL/min/1.73 m^2^, with a median of 68.9 [36.5; 95.9] mL/min/1.73 m^2^. Among patients with eGFR < 60 mL/min/1.73 m^2^ (*n* = 161; 46.3%), the mean eGFR was 32.1 ± 17.3 mL/min/1.73 m^2^ and the median was 31.9 [17.2; 46.4] mL/min/1.73 m^2^. In contrast, patients with eGFR ≥ 60 mL/min/1.73 m^2^ (n = 198; 53.7%) had a mean eGFR of 92.3 ± 16.3 mL/min/1.73 m^2^ and a median of 94.1 [79.0; 103.8] mL/min/1.73 m^2^ (Figure 1).

### 3.1. Demographic Characteristics

The cohort had a mean age of 59.3 years and was predominantly male (57.4%) (Table 1). The most prevalent comorbidities were systemic arterial hypertension (SAH; 48.6%), diabetes mellitus (DM; 28.4%), cardiovascular disease (14%), and chronic kidney disease (CKD; 13.6%). Solid organ transplantation and immunosuppressive therapy were reported in 15% and 12.3% of patients, respectively. Patients with eGFR < 60 were older and had a higher prevalence of comorbidities (Table 1).

In the total cohort, linear regression analysis identified age, SAH, cardiovascular disease, CKD, immunosuppressive therapy, and a history of solid organ transplantation as significant predictors of eGFR values (Table 2). Patients with eGFR < 60, age and CKD remained significant predictors. Conversely, patients with eGFR ≥ 60, age, SAH, cardiovascular disease, immunosuppressive therapy, and obesity were significantly associated with eGFR (Table 2). No significant differences were observed for sex, DM, cerebrovascular disease, COPD, respiratory comorbidities, neoplasia, smoking, or prior hospitalization (Appendix A).

### 3.2. Vital Signs

When evaluating vital signs in the cohort, patients with eGFR < 60 exhibited lower diastolic blood pressure (DBP), mean arterial pressure (MAP), and heart rate (HR) (Table 3). Although SAH was more prevalent in this group, these patients presented with more compromised hemodynamic parameters at admission, suggesting more severe acute illness.

In the total cohort, systolic blood pressure (SBP), DBP, MAP, and HR were significant predictors of eGFR in the linear regression analysis (Table 4). Among patients with GFR ≥ 60, HR and the shock index emerged as the primary predictive variables, whereas no significant predictive variables were identified in the eGFR < 60 group (Table 4). No significant differences were observed between groups for temperature, respiratory rate, SpO_2_, or the ROX index (Appendix A).

### 3.3. Laboratory Findings

Patients with eGFR < 60 exhibited arterial blood gas abnormalities, including lower bicarbonate levels and a negative base excess, consistent with compensated metabolic acidosis (Table 5). This group also showed lower hemoglobin, hematocrit, lymphocyte, and basophil counts, and higher levels of band neutrophil counts, neutrophil-to-lymphocyte ratio, platelet-to-lymphocyte ratio, creatinine, urea, potassium, and D-dimer (Table 5).

In the total cohort, the main predictors of eGFR values included arterial blood gas parameters (pCO_2_, bicarbonate, and base excess), hemoglobin, hematocrit, basophils, lymphocytes, neutrophil-to-lymphocyte ratio, platelet-to-lymphocyte ratio, urea, creatinine, potassium, and D-dimer (Table 6). In the eGFR < 60 group, key predictors were arterial blood gas parameters (pH, bicarbonate, and base excess), hemoglobin, hematocrit, urea, creatinine, urea-to-creatinine ratio, and potassium (Table 6). Among patients with eGFR ≥ 60, the primary predictors were bicarbonate and base excess, urea, creatinine, and potassium (Table 6). No significant associations were observed for pO_2_, lactate, arterial blood glucose, leukocytes, band neutrophils, neutrophils, atypical lymphocytes, monocytes, platelets, C-reactive protein, sodium, or ALT (Appendix A).

### 3.4. Clinical Outcomes

The overall in-hospital mortality rate was 29.2%. A total of 19.8% required at least one session of hemodialysis, 43.7% were admitted to the ICU, and 29% required mechanical ventilation (Table 7).

### 3.5. Predictive Value of eGFR

Next, we performed multivariate regression analyses for all three groups, incorporating variables with a *p*-value < 0.1 from the univariate regressions (Table 2, Table 4 and Table 6). In the total cohort, the model achieved an adjusted R^2^ of 0.756 (*p* < 0.001). For the eGFR < 60 group, the final model produced an adjusted R^2^ of 0.711 (*p* < 0.001), while the eGFR ≥ 60 group yielded an adjusted R^2^ of 0.850 (*p* < 0.001). The predicted values from these models were compared with eGFR values for outcomes predicted from ROC curve analysis and DeLong’s test, as illustrated for mortality (Figure 2A–C), hemodialysis requirement (Figure 3A–C), transfer to the ICU (Figure 4A–C), and need for mechanical ventilation (Figure 5A–C).

In the eGFR < 60 group, based on the results of DeLong’s test, eGFR alone demonstrated significantly better discriminatory ability for predicting the need for hemodialysis (AUC = 0.709) compared to the predicted values from multivariate regression (AUC = 0.651, *p* = 0.039) (Figure 3B). For all other comparisons, there were no significant differences between the eGFR values and the predicted values in discriminating outcomes. Notably, even in the absence of a formal CKD diagnosis, patients with reduced eGFR at admission had a significantly higher risk of acute deterioration requiring dialysis, supporting the prognostic value of eGFR at presentation.

## 4. Discussion

In this study, patients with an eGFR < 60 mL/min/1.73 m^2^ were significantly older and had higher rates of SAH, cardiovascular disease, immunosuppressive therapy, CKD, and solid organ transplantation. Patients in this group exhibited more severe profiles at admission, including lower diastolic blood pressure and mean arterial pressure and reduced heart rate. There was no difference in sex distribution between this group and the rest of the patients. 

Laboratory findings in the lower eGFR group were indicative of metabolic acidosis, anemia, lymphopenia, elevated inflammatory and thrombogenic markers, and hyperkalemia. These abnormalities likely contributed to the significant higher incidence of adverse outcomes—namely mortality, hemodialysis, ICU admission, and mechanical ventilation—observed in this group.

Consistent with our findings, reduced eGFR has been associated with an increased risk for severe COVID-19 (odds ratio [OR]: 2.73), hospitalization (OR: 2.36), and susceptibility to infection (OR: 1.24) [19]. When comparing patients with eGFR in category 1 to those in category 2, the latter had a higher risk of severe COVID-19 [20]. Furthermore, eGFR was inversely associated with the rate of COVID-19 diagnoses, with hazard ratios (HR) of 1.13 for category 2, 1.26 for category 3a, 1.68 for category 3b, and 3.33 for category 4 [21]. Similarly, for 60-day mortality or severe COVID-19, inverse associations were observed: 13.9% (category 1), 16.1% (category 2), 17.8% (category 3a), 22.6% (category 3b), and 23.6% (category 4).

Patients in eGFR category 4 demonstrated the highest risk among all CKD categories [22]. The underlying mechanisms involve direct tubular cytotoxicity to renal tubular cells, glomerular injury in individuals with high-risk *APOL1* genotypes, endothelial damage, renal microthrombi, and inflammatory activation [15]. Cardiorenal syndrome may further exacerbate kidney dysfunction and increase COVID-19 progression and mortality [23].

Moreover, eGFR < 60 is commonly associated with advanced age and poor cardiometabolic profiles, including diabetes and hypertension [24], both of which are known to exacerbate the clinical burden of COVID-19. In this context, hypercoagulability, immune dysregulation, and chronic inflammation may contribute to greater disease severity [25], potentially explaining our findings in patients with reduced eGFR. These comorbidities also increase tissue susceptibility to SARS-CoV-2 infection [26].

SARS-CoV-2 triggers excessive neutrophil activation, resulting in heightened cytokine release, oxidative stress, and inflammation. These processes contribute to endothelial damage and thrombotic complications. Neutrophil extracellular trap formation (NETosis) is strongly associated with poor outcomes in COVID-19 patients [27].

In our population with eGFR < 60, elevated band neutrophil counts and reduced lymphocyte levels likely reflect a higher SARS-CoV-2 viral burden and are indicative of worse clinical outcomes. This observation aligns with previous studies linking increased neutrophil-to-lymphocyte ratios to COVID-19 progression and mortality [28,29]. Transcriptomic analyses further support these associations, demonstrating overexpression of innate immune genes, elevated segmented neutrophil counts, and higher neutrophil-to-lymphocyte ratios—particularly in older individuals with severe disease [30]. These findings underscore the potential utility of immune cell profiles for risk stratification in COVID-19 patients.

Lymphopenia has been consistently associated with severe clinical outcomes in COVID-19, including increased mortality, ICU admission, and the development of acute distress disease syndrome (ARDS) [31]. Its pathogenesis is multifactorial, involving direct viral infection of lymphocytes, cytokine-induced suppression and apoptosis—particularly mediated by IL-6 and TNF-α—bone marrow suppression, and redistribution of lymphocyte to inflamed tissues [32]. Moreover, sustained immune activation can result in lymphocyte exhaustion, leading to both quantitative and functional impairment [33].

Similarly, reduced basophil counts have been frequently observed during the acute phase of COVID-19, in contrast to the recovery phase, and may serve as a marker of disease severity [34]. Basophils act as antigen-presenting cells and enhance humoral immune responses by inducing Th2 lymphocyte differentiation. Consequently, basophil counts are associated with immune dysregulation and an overactive inflammatory response in COVID-19. Monitoring basophil counts, along with other hematologic parameters such as lymphocyte levels and the neutrophil-to-lymphocyte ratio, could provide valuable insights into disease severity and progression [35], particularly in patients with low eGFR.

Other hematological findings in our study included a higher platelet-to-lymphocyte ratio and lower hemoglobin and hematocrit levels in the eGFR < 60 group.

In addition to their role in homeostasis [36], platelets actively participate in innate and adaptive immune responses by recognizing pathogens and promoting immunothrombosis. In COVID-19, SARS-CoV-2 has been detected within platelets, triggering cell death pathways and the release of extracellular vesicles [37]. These processes contribute to platelet dysfunction, characterized by increased activation and aggregation. In our cohort, an elevated platelet-to-lymphocyte ratio in the eGFR < 60 group was associated with greater morbidity and mortality, supporting the utility of platelet-to-lymphocyte ratio as a predictor of poor outcomes [38]. Furthermore, elevated D-dimer levels in this group reflected enhanced thrombotic activity, consistent with previous findings [39,40].

Anemia, frequently observed in patients with eGFR < 60, may arise from chronic inflammation, iron deficiency, and/or vitamin deficiencies [41]. Hemoglobin levels are influenced by multiple factors, including erythrocyte sedimentation rate, serum cholinesterase, ferritin and protein concentrations, the number of chronic diseases, and patient age [41]. A recent meta-analysis also found that anemia prevalence increases with age and the presence of chronic kidney disease [42], aligning with our findings in the eGFR < 60 group. Moreover, anemia has been linked to increased short-term mortality.

These observations highlight the complex interplay between systemic inflammation, cytokine storm, coagulopathy and microthrombus formation in COVID-19 [36]. Together, these processes may lead to intravascular hemolysis and the release of free hemoglobin, which in turn promotes oxidative stress and endothelial damage.

Anemia may exacerbate the pro-thrombotic state observed in COVID-19 by increasing blood viscosity and reducing microcirculatory flow, thereby promoting clot formation. This can further impair microvascular perfusion and contribute to organ dysfunction, particularly in the lungs and kidneys. Additionally, reduced hemoglobin levels compromise oxygen delivery, intensifying hypoxemia and tissue hypoxia—especially in severe cases characterized by respiratory failure. This, in turn, contributes to an increased burden of comorbidities such as cardiovascular and kidney diseases [43].

These risk factors—including hematologic abnormalities and heightened inflammation—were further amplified in patients with eGFR < 60, who also exhibited a higher prevalence of comorbidities, notably solid organ transplantation and chronic immunosuppressive therapy. These findings are consistent with previous studies reporting higher rates of COVID-19 progression and mortality in similar populations [44,45,46,47]. Chronic immunosuppressive therapy may also impair lymphocyte counts and functionality [48], compounding the immune dysregulation observed in these patients.

Arterial blood gas analysis serves as a valuable point-of-care tool for assessing disease severity in COVID-19 [49]. In our study, the most common acid–base disturbance was respiratory alkalosis with secondary metabolic acidosis, consistent with the primary abnormalities reported in hospitalized COVID-19 patients [50]. Moreover, kidney dysfunction in the eGFR < 60 cohort was associated with metabolic acidosis, elevated urea levels, and hyperkalemia.

In conclusion, our findings indicate that an eGFR < 60 mL/min/1.73 m^2^ at hospital admission is associated with an increased likelihood of adverse outcomes in COVID-19. Notably, we demonstrated that a single eGFR measurement at admission serves as a strong and independent predictor of clinical deterioration.

This association provides important insights into disease progression and supports early therapeutic decision-making and risk stratification during future pandemic waves of the COVID-19 pandemic, as well as in other respiratory virus outbreaks.

Additionally, our stratified analysis revealed that admission eGFR alone exhibited predictive performance comparable to that of more complex multivariate models for key clinical outcomes, as confirmed by ROC curve analysis and DeLong’s test. These findings suggest that eGFR could serve as a practical, accessible, and cost-effective tool for early risk assessment in pandemic scenarios.

Importantly, this study was conducted in a public tertiary hospital in Brazil during the first wave of the pandemic, contributing valuable data from a setting underrepresented in the global literature.

Taken together, these findings support the integration of admission eGFR with other clinical and laboratory parameters—including demographic characteristics, vital signals, hematologic indices, arterial blood gas profiles, and inflammatory markers—to provide a more comprehensive understanding of patient risk and pathophysiology.

### Limitations

This study has several limitations. First, our analysis was based on a single eGFR measurement at admission, which may introduce misclassification or measurement bias and does not account for dynamic changes in kidney function. Second, this study was conducted at a single center, limiting the generalizability of the findings. Third, due to constraints in data availability during the early phase of the pandemic, we used admission eGFR < 60 mL/min/1.73 m^2^ as a surrogate marker of kidney impairment, which may include undiagnosed CKD and AKI cases. Additionally, data on proteinuria and albuminuria were not consistently available and thus could not be incorporated into our analysis. Finally, this study was conducted prior to the widespread use of corticosteroids and the availability of COVID-19 vaccines, both of which may influence clinical outcomes. Therefore, further multicenter studies with longitudinal follow-up are needed to validate our findings in more diverse and contemporary contexts.

## Figures and Tables

**Figure 1 idr-17-00079-f001:**
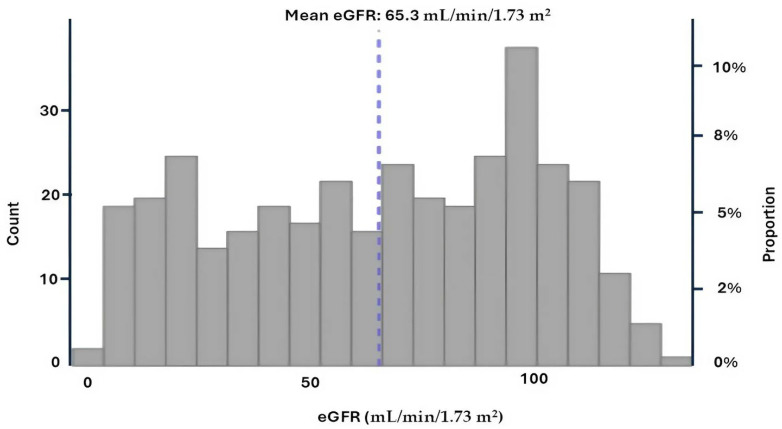
Estimated glomerular filtration rate (eGFR) distribution at admission.

**Figure 2 idr-17-00079-f002:**
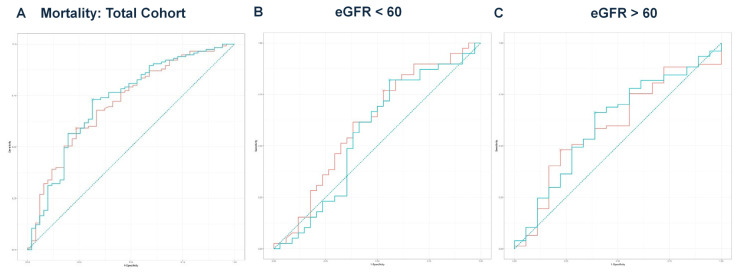
(**A**) ROC curve analysis for mortality in the total cohort. The eGFR had an AUC of 0.715, while the model achieved an AUC of 0.722. DeLong’s test revealed no significant difference (Z = −0.363, *p* = 0.716). The teal curve represents the predicted values (AUC = 0.722 [0.635–0.810], *p* < 0.001), and the orange curve represents eGFR (AUC = 0.715 [0.629–0.800], *p* < 0.001). (**B**) ROC curve analysis for mortality in the eGFR < 60 group. The eGFR value had an AUC of 0.613, while the predicted model had an AUC of 0.563, with no significant difference (Z = 1.816, *p* = 0.069). The teal curve represents the predicted values (AUC = 0.563 [0.425–0.702], *p* < 0.001), and the orange curve represents eGFR (AUC = 0.613 [0.480–0.746], *p* < 0.001). (**C**) ROC curve analysis for mortality in the eGFR ≥ 60 group. The ROC curve for eGFR yielded an AUC of 0.596 compared to the predicted model’s AUC of 0.612. DeLong’s test showed no significant difference (Z = −0.592, *p* = 0.554). The teal curve represents the predicted values (AUC = 0.612 [0.463–0.761], *p* < 0.001), and the orange curve represents eGFR (AUC = 0.596 [0.449–0.743], *p* < 0.001).

**Figure 3 idr-17-00079-f003:**
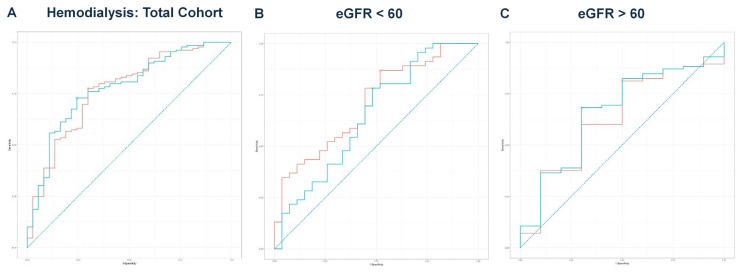
(**A**) ROC curve analysis for hemodialysis in the total cohort. The AUC for eGFR was 0.766, while the AUC for the predicted values was 0.776. DeLong’s test indicated no significant difference (Z = −0.502, *p* = 0.616). The teal curve represents the predicted values (AUC = 0.776 [0.691–0.861], *p* < 0.001), and the orange curve represents eGFR (AUC = 0.766 [0.677–0.855], *p* < 0.001). (**B**) ROC curve analysis for hemodialysis in the eGFR < 60 group. A comparison of eGFR and predicted values resulted in DeLong’s test statistic of Z = 2.069 (*p* = 0.039), with AUC values of 0.709 and 0.651, respectively. The teal curve represents the predicted values (AUC = 0.651 [0.513–0.788], *p* < 0.001), and the orange curve represents eGFR (AUC = 0.709 [0.586–0.831], *p* < 0.001). (**C**) ROC curve analysis for hemodialysis in the eGFR ≥ 60 group. The ROC curve analysis produced AUC values of 0.631 for eGFR and 0.659 for the predicted values. DeLong’s test showed no significant difference (Z = −1.262, *p* = 0.207). The teal curve represents the predicted values (AUC = 0.659 [0.475–0.842], *p* < 0.001), and the orange curve represents eGFR (AUC = 0.631 [0.446–0.816], *p* < 0.001).

**Figure 4 idr-17-00079-f004:**
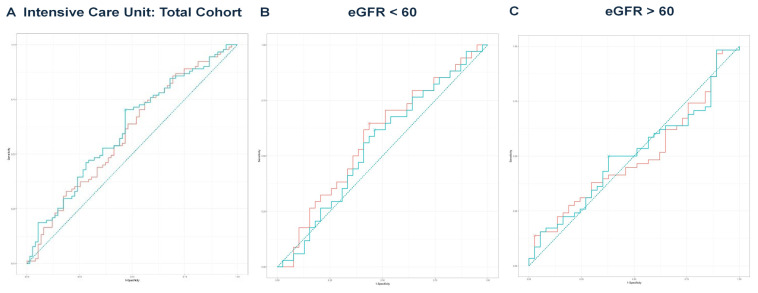
(**A**) ROC curve analysis for ICU in the total cohort. The predicted model had an AUC of 0.613, compared to an AUC of 0.595 for eGFR. DeLong’s test was non-significant (Z = −1.019, *p* = 0.308). The teal curve represents the predicted values (AUC = 0.613 [0.527–0.699], *p* < 0.001), and the orange curve represents eGFR (AUC = 0.595 [0.508–0.682], *p* < 0.001). (**B**) ROC curve analysis for ICU in the eGFR < 60 group. The predicted model yielded an AUC of 0.551, while eGFR had an AUC of 0.583. DeLong’s test indicated no significance (Z = 1.194, *p* = 0.232). The teal curve represents the predicted values (AUC = 0.551 [0.417–0.684], *p* < 0.001), and the orange curve represents eGFR (AUC = 0.583 [0.450–0.716], *p* < 0.001). (**C**) ROC curve analysis for ICU in the eGFR ≥ 60 group. The predicted model’s AUC was 0.507, compared to an eGFR AUC of 0.496. DeLong’s test showed no significant difference (Z = −0.024, *p* = 0.981). The teal curve represents the predicted values (AUC = 0.507 [0.388–0.626], *p* < 0.001), and the orange curve represents eGFR (AUC = 0.496 [0.385–0.623], *p* < 0.001).

**Figure 5 idr-17-00079-f005:**
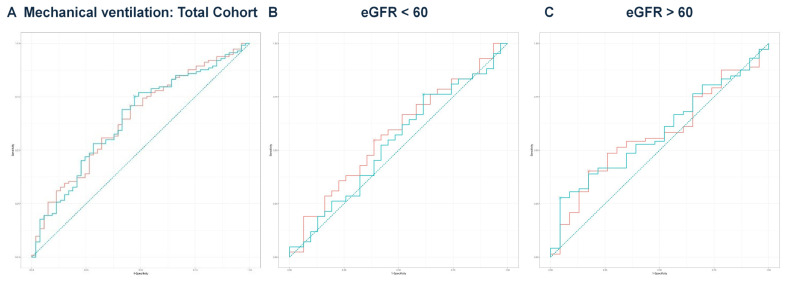
(**A**) ROC curve analysis for mechanical ventilation in the total cohort. ROC curve analysis showed no significant difference in DeLong’s test (Z = 0.097, *p* = 0.923). The predicted model had an AUC of 0.646, while eGFR had an AUC of 0.647. The teal curve represents the predicted values (AUC = 0.646 [0.555–0.736], *p* < 0.001), and the orange curve represents eGFR (AUC = 0.647 [0.558–0.737], *p* < 0.001). (**B**) ROC curve analysis for mechanical ventilation in the eGFR < 60 group. The predicted model had an AUC of 0.537 compared to an eGFR AUC of 0.576. DeLong’s test was non-significant (Z = 1.486, *p* = 0.137). The teal curve represents the predicted values (AUC = 0.537 [0.401–0.673], *p* < 0.001), and the orange curve represents eGFR (AUC = 0.576 [0.442–0.710], *p* < 0.001). (**C**) ROC curve analysis for mechanical ventilation in the eGFR ≥ 60 group. The predicted model’s AUC was 0.585, while eGFR had an AUC of 0.576. DeLong’s test revealed no significant difference (Z = −0.387, *p* = 0.699). The teal curve represents the predicted values (AUC = 0.585 [0.460–0.711], *p* < 0.001), and the orange curve represents eGFR (AUC = 0.576 [0.448–0.704], *p* < 0.001).

**Table 1 idr-17-00079-t001:** Demographic data according to eGFR values.

Variables	Total (n = 359)	eGFR < 60 (n = 161)	eGFR ≥ 60 (n = 198)	Statistical Test	*p*-Value
Female (%)	42.6	47.2	37.9	3.169	0.075
Male (%)	57.4	52.8	62.1		
Age (years)	60 [48.5; 70.0]	64 [53.0; 74.0]	58 [46; 67.7]	−3.121	0.002
SAH (%)	48.6	57.1	41.6	8.541	0.003
DM (%)	28.4	29.8	27.3	0.282	0.595
CKD (%)	13.6	25.5	4.0	35.580	<0.001
Solid organ transplantation (%)	15.0	25.5	6.6	24.820	<0.001
Immunosuppression (%)	12.3	18.0	7.6	8.994	0.003
Cardiovascular disease (%)	14.0	17.4	11.2	2.856	0.091
Cerebrovascular disease (%)	3.9	3.1	4.5	0.419	0.483
COPD (%)	2.5	1.9	3.0	0.495	0.482
Asthma (%)	3.3	1.9	4.5	1.977	0.160
Other respiratory illness (%)	2.5	2.5	2.5	0.006	0.980
Neoplasia (%)	6.1	8.1	4.5	1.966	0.161
Obesity (%)	11.4	9.3	13.1	1.277	0.258
Smoking (%)	8.6	11.2	6.6	2.397	0.122
Previous hospitalization (%)	4.5	4.3	4.5	0.008	0.928

SAH = systemic arterial hypertension; DM = diabetes mellitus; CKD = chronic kidney disease; COPD = chronic obstructive pulmonary disease. Age is reported as the median and interquartile range [Q1; Q3].

**Table 2 idr-17-00079-t002:** Linear regression of GFR on demographic data.

Variables	Total (n = 359)			eGFR < 60 (n = 161)			eGFR ≥ 60 (n = 198)		
	Adjusted R^2^	F-Test	*p*-Value	Adjusted R^2^	F-Test	*p*-Value	Adjusted R^2^	F-Test	*p*-Value
Age	0.042	16.830	<0.001	0.047	8.805	0.003	0.249	66.160	<0.001
SAH	0.028	11.330	<0.001	0.009	2.404	0.123	0.065	14.670	<0.001
Cardiovascular disease	0.014	6.034	0.015	−0.006	0.087	0.768	0.032	7.402	0.007
CKD	0.152	65.190	0.001	0.136	26.240	<0.001	0.004	1.811	0.180
Immunosuppression	0.037	14.660	<0.001	0.004	1.637	0.203	0.021	5.138	0.025
Solid organ transplantation	0.075	30.130	<0.001	0.005	1.739	0.189	0.008	2.606	0.108
Obesity	−0.002	0.108	0.743	0.001	1.239	0.267	0.029	6.932	0.009

SAH = systemic arterial hypertension; CKD = chronic kidney disease.

**Table 3 idr-17-00079-t003:** Vital signs according to eGFR values.

Variables	Total (n = 359)	eGFR < 60 (n = 161)	eGFR ≥ 60 (n = 198)	*t*-Test	*p*-Value
SBP (mmHg)	128 [115; 143.2]	126 [110; 144.5]	129 [118; 143]	1.304	0.193
DBP (mmHg)	79 [70; 89]	75 [65.7; 84]	80 [71; 90.8]	3.144	0.002
MAP (mmHg)	95 [85; 106]	92.5 [81; 103.2]	96 [87; 107]	2.576	0.011
Heart Rate (bpm)	93.5 [81; 109.2]	88.5 [79.00; 104.00]	98.4 [87.00; 111.7]	2.655	0.008
Shock Index	0.74 [0.63; 0.86]	0.73 [0.58; 0.86]	0.75 [0.65; 0.85]	0.042	0.966
Temperature (°C)	36.5 [36.0; 37.0]	36 [36.0; 36.9]	36.6 [36.0; 37.0]	0.119	0.905
Respiratory rate (bpm)	25 [21.0; 29.0]	25 [20.0; 29.0]	24 [22.0; 29.0]	−0.366	0.715
SpO_2_ (%)	93 [89.0; 95.0]	93 [89.0; 95.0]	92 [89.0; 95.0]	0.763	0.446
ROX Index	17.8 [14.9; 20.9]	17.7 [14.9; 21.6]	17.9 [14.8; 20.3]	0.990	0.324

SBP = systolic blood pressure; DBP = diastolic blood pressure; MAP = mean arterial pressure; SpO_2_ = peripheral capillary oxygen saturation; ROX = respiratory rate oxygenation index. All variables are reported as the median and interquartile range [Q1; Q3].

**Table 4 idr-17-00079-t004:** Linear regression analysis of vital signs according to eGFR values.

	Total (n = 359)			eGFR < 60 (n = 161)			eGFR ≥ 60 (n = 198)		
	Adjusted R^2^	F-Test	*p*-Value	Adjusted R^2^	F-Test	*p*-Value	Adjusted R^2^	F-Test	*p*-Value
SBP (mmHg)	0.011	4.596	0.033	0.018	3.451	0.065	−0.001	0.787	0.376
DBP (mmHg)	0.041	14.150	<0.001	0.002	1.317	0.253	0.009	2.619	0.107
MAP (mmHg)	0.032	11.130	<0.001	0.010	2.323	0.130	0.006	2.043	0.155
Heart rate (bpm)	0.036	12.640	<0.001	−0.006	0.188	0.665	0.085	17.140	<0.001
Shock Index	−0.003	0.005	0.945	0.018	3.481	0.064	0.030	6.128	0.014

SBP = systolic blood pressure; DBP = diastolic blood pressure; MAP = mean arterial pressure.

**Table 5 idr-17-00079-t005:** Laboratory data according to eGFR levels.

	Total (n = 359)	eGFR < 60 (n = 161)	eGFR ≥ 60 (n = 198)	*t*-Test	*p*-Value
pH	7.44 [7.39; 7.47]	7.40 [7.35; 7.45]	7.46 [7.42; 7.48]	1.924	0.056
pCO_2_ (mmHg)	32.3 [28.1; 36.3]	31.9 [26.0; 35.7]	32.5 [29.2; 36.4]	1.585	0.114
pO_2_ (mmHg)	62.9 [53.4; 73.4]	63.3 [53.0; 79.3]	62.7 [53.8; 71.2]	−0.496	0.620
Bicarbonate (mEq/L)	21.3 [18.5; 23.5]	19.3 [16.6; 22.6]	22.4 [20.8; 24.7]	7.117	<0.001
Base excess	−1.7 [−4.6; 0.8]	−4.1 [−7.7; −1.6]	−0.3 [−1.9; 1.7]	8.057	<0.001
Lactate (mg/dL)	13.2 [9.0; 19.0]	13.0 [9.0; 19.0]	14.0 [10.0; 19.0]	−0.457	0.648
Arterial blood glucose (mg/dL)	135.0 [112.0; 196.0]	129.0 [110.0; 187.0]	139.5 [113.0; 201.7]	−0.008	0.994
Hemoglobin (g/dL)	13.5 [12.0; 14.6]	12.7 [10.8; 14.1]	13.9 [12.7; 15.0]	5.603	<0.001
Hematocrit (%)	39.9 [36.0; 43.3]	38.2 [31.9; 42.0]	40.8 [37.6; 44.1]	4.811	<0.001
Leukocytes (/µL)	7610 [5270; 10,280]	7720 [5330; 10,280]	7505 [5212; 10,267]	−0.750	0.454
Band neutrophils (/µL)	0 [0; 233]	0 [0; 362]	0 [0; 173]	−2.107	0.036
Neutrophils (/µL)	5862 [3771; 8164]	5978 [3939; 8016]	5776 [3727; 8297]	−0.954	0.341
Eosinophils (/µL)	4 [0; 33]	2 [0; 33]	4 [0; 33]	0.692	0.489
Basophils (/µL)	9 [0; 21]	6 [0; 20]	11 [0; 24]	2.099	0.037
Lymphocytes (/µL)	995 [643; 1375]	858 [507; 1291]	1029 [719; 1487]	2.252	0.025
Atypical lymphocytes (/µL)	0 [0;0]	0 [0;0]	0 [0;0]	0.155	0.877
Monocytes (/µL)	437 [283; 649]	449 [275; 685]	434 [287; 640]	−0.475	0.635
Neutrophil-to-lymphocyte ratio	6.3 [3.6; 9.8]	7.1 [4.0; 12.1]	5.2 [3.4; 8.6]	−2.723	0.007
Platelets (/µL), ×10^3^	189 [148; 239]	183.5 [131.5; 232.5]	191 [157; 242.5]	0.875	0.382
Platelets-to-lymphocytes ratio	189.4 [125.8; 304.4]	210.7 [128.4; 345.1]	183.0 [125.0; 256.1]	−2.566	0.011
C-Reactive protein (mg/L)	94.3 [52.2; 179.3]	84.0 [47.8; 177.8]	96.6 [53.8; 183.6]	−0.154	0.878
Urea (mg/dL)	41.5 [28.0; 73.0]	78.0 [56.0; 117.0]	29.0 [24.0; 37.0]	13.758	<0.001
Creatinine (mg/dL)	1.09 [0.84; 1.95]	2.13 [1.47; 3.42]	0.86 [0.72; 1.00]	−9.024	<0.001
Urea-to-creatinine ratio	34.5 [27.7; 43.6]	34.3 [26.9; 44.2]	34.5 [28.0; 43.1]	0.490	0.624
Sodium (mEq/L)	136 [132; 139]	136 [131; 139]	136 [133; 140]	1.598	0.111
Potassium (mEq/L)	4.5 [4.0; 4.9]	4.8 [4.4; 5.5]	4.2 [3.9; 4.6]	−7.262	<0.001
ALT (U/L)	29 [18; 49]	23.5 [15; 39]	33 [23; 55]	−0.521	0.603
D-Dimer (µg/mL)	1.3 [0.8; 2.2]	1.6 [0.9; 2.5]	1.1 [0.6; 1.8]	−2.601	0.010

pCO_2_ = partial pressure of carbon dioxide; pO_2_ = partial pressure of oxygen; ALT = alanine aminotransferase. All variables are reported as the median and interquartile range [Q1; Q3].

**Table 6 idr-17-00079-t006:** Linear regression analysis of laboratory parameters according to eGFR values.

	Total (n = 369)			eGFR < 60 (n = 161)			eGFR ≥ 60 (n = 198)		
	Adjusted R^2^	F-Test	*p*-Value	Adjusted R^2^	F-Test	*p*-Value	Adjusted R^2^	F-Test	*p*-Value
pH	0.009	3.768	0.053	0.198	34.070	<0.001	0.000	1.027	0.312
pCO_2_ (mmHg)	0.014	5.197	0.023	−0.001	0.914	0.341	0.010	2.621	0.108
Bicarbonate (mEq/L)	0.225	85.560	<0.001	0.149	24.400	<0.001	0.034	6.450	0.012
Base excess	0.273	109.600	<0.001	0.228	40.480	<0.001	0.029	5.629	0.019
Hemoglobin (g/dL)	0.100	40.500	<0.001	0.118	22.310	<0.001	0.000	0.915	0.340
Hematocrit (%)	0.078	30.890	<0.001	0.107	20.040	<0.001	0.001	1.273	0.261
Eosinophils (/µL)	−0.001	0.482	0.488	0.027	5.436	0.021	0.012	3.301	0.071
Basophils (/µL)	0.013	5.631	0.018	−0.006	0.100	0.753	−0.005	0.009	0.893
Lymphocytes (/µL)	0.020	8.220	0.004	0.009	2.464	0.118	−0.002	0.656	0.419
Neutrophil-to-lymphocyte ratio	0.015	6.247	0.013	−0.005	0.202	0.654	−0.005	0.054	0.817
Platelet-to-lymphocyte ratio	0.011	4.946	0.027	−0.003	0.461	0.498	−0.003	0.385	0.536
Urea (mg/dL)	0.561	456.900	<0.001	0.444	128.900	<0.001	0.245	64.450	<0.001
Creatinine (mg/dL)	0.387	226.700	<0.001	0.450	131.700	<0.001	0.487	187.700	<0.001
Urea-to-creatinine ratio	0.006	3.029	0.083	0.156	30.610	<0.001	0.006	2.214	0.138
Potassium (mEq/L)	0.195	84.830	<0.001	0.058	10.420	0.002	0.059	12.980	<0.001
D-Dimer (µg/mL)	0.047	13.360	<0.001	0.023	3.688	0.057	0.015	3.148	0.078

pCO_2_ = partial pressure of carbon dioxide.

**Table 7 idr-17-00079-t007:** COVID-19 related outcomes according to eGFR values.

	Total (n = 359)	eGFR < 60 (n = 161)	eGFR ≥ 60 (n = 198)	χ^2^	*p*-Value
Mortality (%)	29.2	41.6	19.2	21.6	<0.001
Hemodialysis (%)	18.8	32.3	9.6	28.8	<0.001
Intensive care unit (%)	43.1	50.9	37.9	6.1	0.013
Mechanical ventilation (%)	29.0	39.8	21.2	14.7	<0.001

## Data Availability

The data presented in this study are available upon request from the corresponding authors. The data are not publicly available due to clinical patient information.

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
