# Peer review of "The Role of Kidney Function in Predicting COVID-19 Severity and Clinical Outcomes: A Retrospective Analysis"

_2036-7449, 2025, doi:10.3390/idr17040079_

Round 1

Reviewer 1 Report (Previous Reviewer 1)

Comments and Suggestions for Authors

The manuscript provides a clear and well-conducted retrospective analysis of 359 hospitalised COVID-19 patients examining the relationship between eGFR on admission and clinical outcomes. The study is well structured, clearly written and uses appropriate statistical methods. However, the main findings that lower eGFR is associated with higher mortality, higher need for CRT and more ICU admissions are already well established in the literature. Several large studies and meta-analyses have already shown that both pre-existing CKD and reduced eGFR on admission are strong, independent predictors of poor outcomes in COVID-19 patients. Therefore, the results largely confirm what is already known in this field.

Minor comments:

The introduction and discussion should emphasize how this study adds to the existing body of evidence, and more explicitly acknowledge the well-established nature of the main findings.

Author Response

We thank the reviewers for their thoughtful and constructive comments, which have helped us improve the clarity and relevance of our manuscript. Regarding the English editing, we have carefully revised the manuscript, and all changes are highlighted in yellow in the revised version.  We also improved the image quality of Figure 1.

Reviewer 1:

Comment:
The manuscript is clear and well-structured, but the main findings (lower eGFR associated with worse COVID-19 outcomes) are already well-established. Please emphasize how this study adds to the existing literature and acknowledge that these findings are already known.

Response:
We appreciate this valuable observation. While it is true that reduced eGFR and pre-existing CKD are known risk factors for worse COVID-19 outcomes, our study contributes in the following ways:

  1. We specifically analyzed admission eGFR regardless of CKD diagnosis, capturing acute or previously undiagnosed renal dysfunction.
  2. Our stratified analysis demonstrates that admission eGFR alone has comparable predictive value to more complex multivariate models, especially for dialysis requirement (as shown by ROC and DeLong’s test results).
  3. The study was conducted in a public tertiary hospital in Brazil during the first wave, a setting underrepresented in the global literature.
  4. The manuscript includes detailed associations between eGFR and vital signs, inflammatory markers, gas exchange parameters, and hematologic alterations, providing a broader pathophysiological context.

To address the reviewer’s comment, we revised the Introduction and Discussion to better situate our work within the existing literature and to clarify our study’s specific contributions [Changes highlighted in blue in the revised manuscript]. 

Reviewer 2 Report (Previous Reviewer 2)

Comments and Suggestions for Authors

Although the manuscript is not on a completely new topic, it now adds some new and interesting data. It has been sufficiently improved by the authors' thorough revision that I believe it can now be considered suitable for publication in the Journal in its current form.

Author Response

We thank the reviewers for their thoughtful and constructive comments, which have helped us improve the clarity and relevance of our manuscript. Regarding the English editing, we have carefully revised the manuscript, and all changes are highlighted in yellow in the revised version.  We also improved the image quality of Figure 1.

Reviewer 2:

Comment:
The revised version is improved and now suitable for publication.

Response:
We thank the reviewer for the positive feedback and are pleased that the revisions have enhanced the quality and clarity of the manuscript.

Reviewer 3 Report (Previous Reviewer 3)

Comments and Suggestions for Authors

The content of the manuscript improved, most of previous concerns have been addressed - introduction was enriched with references, information on hypertension was clarified.

Limitations contain the information about the eGFR <60ml/min being a surrogate marker of renal impairment of unknown history (AKI or previously undiagnosed CKD).

In order to strengthen the predictive value of this parameter, data on proteinuria/microalbuminuria should be added.

Author Response

We thank the reviewers for their thoughtful and constructive comments, which have helped us improve the clarity and relevance of our manuscript. Regarding the English editing, we have carefully revised the manuscript, and all changes are highlighted in yellow in the revised version.  We also improved the image quality of Figure 1.

Reviewer 3:

Comment:
The manuscript has improved, but including data on proteinuria/microalbuminuria would strengthen the use of eGFR as a predictive parameter.

Response:
We agree that adding proteinuria or microalbuminuria data would improve the interpretation of kidney function. Unfortunately, due to data limitations during the first wave of the pandemic, these measures were not consistently recorded in our cohort and thus could not be included in our analysis.

We now explicitly mention this limitation in the revised Discussion and Limitations sections to transparently address the impact of this missing variable. We also clarify that eGFR < 60 mL/min/1.73 m² served as a surrogate for undifferentiated kidney impairment, encompassing both AKI and undiagnosed CKD.

Sincerely,

Érika B Rangel, MD, PhD

Round 2

Reviewer 1 Report (Previous Reviewer 1)

Comments and Suggestions for Authors

Dear Authors,

Thank you for your patience and for addressing the comments thoroughly. The revisions made are appropriate, and I believe that the current version of the manuscript is suitable for publication.

Wishing you every success with your paper.

This manuscript is a resubmission of an earlier submission. The following is a list of the peer review reports and author responses from that submission.

Round 1

Reviewer 1 Report

Comments and Suggestions for Authors

The manuscript presents a retrospective analysis of the impact of kidney function on COVID-19 outcomes. While the association between reduced eGFR (<60 mL/min/1.73 m²) and poorer clinical outcomes is not novel, it remains clinically relevant and supports the use of eGFR for risk stratification. I have several comments for the authors.

  • Revise the abstract to include specific outcomes and eGFR’s practical value.
  • The study’s novelty isn’t fully highlighted. How does it build on prior work? Please add a sentence on research gaps in this topic.
  • It is unclear how CKD was confirmed. Did the authors rely on admission eGFR, potentially leading to misclassification of AKI as CKD? How did the authors address cases of CKD with preserved kidney function? It should be clearly stated in the Methods.
  • There is a lack of clarity in how the authors interpret the finding that patients with eGFR < 60 had a higher risk of requiring hemodialysis. While this is statistically supported, the statement that “hemodialysis risk is higher in kidney failure patients than in those with normal kidney function” is self-evident and risks sounding tautological.
  • The Results section is too vague and difficult to follow in parts. Important findings are often embedded in long lists of variables or p-values without sufficient context or interpretation. This makes it challenging for readers to grasp the significance or clinical implications of the reported associations. In my view, rather than listing numerous statistical values, the authors should present the key findings using a clear narrative. Consider organizing the results with clearer subheadings and moving some data to a supplementary file.
  • Simplify the Tables or Reference Them Clearly: Readers should not be required to cross-reference multiple dense tables just to understand the main points. Key findings should be highlighted in the text, with tables serving to support, not replace, the narrative.
  • The discussion extensively describes mechanisms (e.g., NETosis, lymphopenia, platelet dysfunction), which feel somewhat detached from the primary data. Please focus on your findings in the Discussion.
  • Replace the term “eGFR stage” with “eGFR category” and “renal failure” with “kidney failure.” Review the manuscript for other outdated terms (e.g., “renal dysfunction,” “renal impairment”) and consider using “kidney function impairment.”

Reviewer 2 Report

Comments and Suggestions for Authors

The study reported in this manuscript does not add any novelty to the previous literature on the topic.

Reviewer 3 Report

Comments and Suggestions for Authors

Even of the topic is not very original, and the fact that kidney function impairment would aggravate the prognosis in Covid patients is quite predictable, hard data on the subject is of clinical value. However, the major methodological concern is about classification of patients with eGFR < 60ml/min on admission. Some of them are indeed diagnosed with CKD, but the majority has no documented previous history of renal impairment. Therefore, their renal function may result from Covid-related AKI or exacerbation of undiagnosed CKD. Moreover, the multiplicity of complications (anemia, acidosis, hyperkalemia, coagulation anomalies, NLR, PLR increase) and their relation to subclinical inflammation suggest that the chronic insufficiency is more probable. Thus, there should be a re-evaluation of kidney function and all abnormal parameters after 3 months so that the primary character  (AKI? AKD? CKD?) of kidney injury is defined. Parathormone values should also be presented. With such follow up, the potential predictive value of eGFR on admission will either be confirmed or contradicted.

Minor:

The abstract contains a statement that patients with eGFR<60 had a higher rate of hypertension, yet in the results there is an information that they had lower blood pressure values than those with eGFR > 60. Please explain.

Introduction lines 48-62 – the Authors list multiple risk factors/complications/markers related to Covid clinical course and severity, but only one reference is given.  Please add other publications on the subject.